# Classification of images using Gaussian copula model in empirical cumulative distribution function space

**Sapto Wahyu Indratno[1], Sri Winarni[2]\*, Kurnia Novita Sari[1]**

**1** Statistics Research Group, Faculty of Mathematics and Natural Sciences, Institut Teknologi Bandung, Bandung West Java, Indonesia, **2** Department of Statistics, Faculty of Mathematics and Natural Sciences, Padjadjaran University, Sumedang, West Java, Indonesia

\* sri.winarni@unpad.ac.id

## Abstract

This study introduces an innovative approach to image classification that uses Gaussian copulas with an Empirical Cumulative Distribution Function (ECDF) approach. The strategic use of distribution functions as feature descriptors simplifies the approach and enables a better understanding of the correlation structure between features in the image. This approach helps the model understand the contextual relationships between different parts of the image, resulting in a more abstract representation than a direct representation of individual pixel values. The proposed model utilizes the Distribution Function of the Distribution Value (DFDV) as the margin distribution. The Modified National Institute of Standards and Technology (MNIST) dataset is comprehensively used to assess the effectiveness of this model. The results show that the model achieves a noteworthy level of accuracy, with an average accuracy of 62.22% and a peak accuracy of 96.92%. This success was obtained by applying the Inference Function for Marginals (IFM) principles during the training stage.

## Introduction

Image classification constitutes a fundamental aspect of computer vision, encompassing the systematic categorization of images into distinct classes or categories based on their discernible visual features and characteristics. The modeling process involves acquiring knowledge through pattern learning from the training data. The program's efficacy in capturing these data patterns is directly contingent upon the magnitude of the training dataset [1]. The practical applications of image classification are diverse, spanning tasks such as object identification [2], facial detection [3], handwriting recognition [4–8], automated automotive number plate recognition [9], and medical condition diagnosis [10]. Nevertheless, image classification presents a formidable challenge due to the necessity of differentiating between objects and patterns with visual similarities. To address this challenge, meaningful features, including shape, texture, color, or a combination thereof, are judiciously extracted from the images [11]. These extracted features are then employed to train a machine learning model, facilitating the acquisition of the capability to discern patterns and make predictions for new, previously unseen

**Data Availability Statement:** The dataset is freely available at https://www.kaggle.com/datasets/hojjatk/mnist-dataset.

**Funding:** This research was funded by the Bandung Institute of Technology International

Research Program in 2023 titled "Development of Deep Gaussian Process". The funders had no role in study design, data collection and analysis, decision to publish, or preparation of the manuscript.

**Competing interests:** The authors have declared that no competing interests exist.

images. Consequently, image classification emerges as a critical component in automating tasks that would otherwise demand human intervention.

Developing a model for image classification is an engaging pursuit, and various methodologies have been formulated to confront this challenge. Prior investigations have scrutinized the identification of image shapes and textures, juxtaposing the Support Vector Machine (SVM) technique against K-Nearest Neighbor (KNN) methods [12]. Subsequent inquiries have expanded into the domain of Support Kernel Machine (SKM), Probabilistic Neural Network (NN) [13, 14], SVM, and Artificial Neural Network (ANN) methodologies [15, 16]. Nevertheless, contemporary strides in image classification techniques have shifted toward deep learning, prominently employing Convolutional Neural Networks (CNN) [17–19]. Despite the encouraging outcomes showcased by CNN, the complexity of the model may introduce practical challenges in its implementation [20–22]. As the complexity of deep learning models increases, interpreting them becomes progressively complex [23–25].

Several studies have examined applying Gaussian copulas as classification models for image analysis. For instance, a study utilized a Gaussian copula model with the Gamma distribution as the marginal distribution to classify image textures in the Vision Texture database [26]. Another investigation introduced a Gaussian copula-based probabilistic classifier to distinguish between object and background images [27]. Additionally, a research endeavor employed the copula distribution function and a diagnostic classifier to categorize brain image data [28]. The application of Gaussian copulas has extended to statistical pattern recognition for acute inflammation data [29] and the classification of hyperspectral images [30]. Several studies incorporated transformations as inputs to the Gaussian copula model [31–33], with all investigations using pixel intensity values as inputs. Although these studies have shown promising results, the reliance on pixel intensity values may pose intensive challenges to extracting more abstract features. Therefore, this study proposes a Gaussian copula with an Empirical Cumulative Distribution Function (ECDF) approach for extracting image features. This study represents an advancement in research focused on image classification using symbolic data [34–36].

This study introduces an effective new approach to image classification by combining the Gaussian copula model and applying the cumulative distribution function strategy. Compared to previous research in image classification using Gaussian copula models, this approach is a step forward that aims to simplify the methodology by exploiting the distribution of pixels in an image to describe its characteristics accurately. This approach helps the model understand the contextual relationships between elements in the image, resulting in a more abstract representation of the image than that obtained from individual pixel values. In addition, by using distributions as features, memory to store distribution parameters is much more efficient than the memory required to directly store many image pixel values.

The research methodology systematically divides each image into $p$ different partitions, where the Empirical Cumulative Distribution Function (ECDF) is computed within each partition, generating a set of ECDF functions for $N$ images. At a specified point $T_t$ the ECDF values from each function are amalgamated to construct a Distribution Function of Distribution Values (DFDV), effectively encapsulating the distinct characteristics of the image object. Selecting s points in one partition yields $s$ points are taken in one partition, this results in $s$ points in one partition yields s functions to comprehensively represent the image. These partitions originate from the same image, so the interdependence is modeled using a Gaussian copula joint distribution function. The primary objective of this research is to establish a classification model grounded in the Gaussian copula, employing a cumulative distribution function space approach. This strategy aims to enhance the clarity and efficiency of image

classification by meticulously analyzing distribution functions derived from partitioned image data.

## Research methodology

### The Gaussian copula model using ECDF and DFDV

Suppose there are $N$ images denoted by $X_1, X_2, \ldots, X_N$, where $X_i \in \{0, 1, 2, \ldots, 255\}$, $i = 1, 2, \ldots, N$. Each of these images is partitioned into $p$ equal sized-sections. Let $X_{j,i}$ represent random variables indicating pixel values in partition $j$ and image $i$, where $j = 1,2,\ldots,p$ and $i = 1,2,\ldots,N$. For each partition an empirical cumulative distribution of the pixel values can be derived as follows:

$$\mathcal{F}_{j,i}(x) = \frac{M_{j,i}(x)}{\text{total number of pixels in the } j-\text{th partition}} \tag{1}$$

where $M_{j,i}(x)$ is the count of pixel values in partition $j$ of image $i$ that are less than or equal to $x$. Therefore, one gets the following collection of empirical cumulative distribution of the partition $j$:

$$\mathfrak{F}^j(.) = \{\mathcal{F}_{j,1}(.), \mathcal{F}_{j,2}(.), \ldots, \mathcal{F}_{j,N}(.)\} \text{ with } j = 1, 2, \ldots, p. \tag{2}$$

Note that $\mathcal{F}_{j,i}(x) \in [0,1]$ for all $x$.

The collection of empirical cumulative distribution allows us to investigate the character of partition $j$ further. Let us consider specific points $T_1^j, T_2^j, \ldots, T_{n_j}^j$, where $T_k^j \in \{0, 1, \ldots, 255\}$. From (2) one gets $\mathfrak{F}^j(T_k^j) = \{\mathcal{F}_{j,1}(T_k^j), \mathcal{F}_{j,2}(T_k^j), \ldots, \mathcal{F}_{j,N}(T_k^j)\}$, $k = 1, 2, \ldots, n_j$. Therefore, one can create a mapping from the interval [0,1] to [0,1] as follows:

$$G_{T_k^j}^j(y) = P(\{\mathcal{F}_{j,i} \in \mathfrak{F}^j | \mathcal{F}_{j,i}(T_k^j) \leq y\}) \tag{3}$$

Here, $\mathcal{F}_{j,i}(T)$ represents the value of the ECDF for partition $j$ of image $i$ at point $T$ [36]. Since $G_{T_k^j}^j(y)$ is a cumulative distribution of empirical cumulative distribution values at $T_k^j$, it is called as Distribution Function of Distribution value (DFDV). Therefore, for partition $j$ one gets $n_j$ DFDV, $j = 1,2,\ldots,p$. To include dependence of the $p$ partitions, we construct a joint cummulative distribution as follows:

$$H_T(y_1^1, \ldots, y_{n_p}^p) = P(\mathcal{F}_j \in \mathfrak{F} | \mathcal{F}_{1,j}(T_1^1) \leq y_1^1, \ldots, \mathcal{F}_{p,j}(T_{n_p}^p) \leq y_{n_p}^p, j = 1, \ldots, N) \tag{4}$$

where $\mathfrak{F} = (\mathcal{F}_1, \ldots, \mathcal{F}_N), \mathcal{F}_j = (\mathcal{F}_{1,j}, \ldots, \mathcal{F}_{p,j})$ and $T = ((T_1^1, \ldots, T_{n_1}^1), \ldots, (T_1^p, \ldots, T_{n_p}^p))$.

To model above joint cumulative distribution, we use copula model. A copula is a mathematical function that describes the dependence structure between random variables, enabling the modeling of their marginal distribution dependencies [37–41]. By utilizing the distribution functions $G_{T_1^1}, \ldots, G_{T_1^2}, \ldots, G_{T_{n_p}^p}$ which specify the distribution of distribution values at points $T_1^1, \ldots, T_1^2, \ldots, T_{n_p}^p$, and the joint distribution function $H_T$, we can formulate a copula function for all $(y_1^1, \ldots, y_{n_p}^p)$ in the following manner:

$$H_T(y_1^1, \ldots, y_{n_p}^p) = C(G_{T_1^1}(y_1^1), \ldots, G_{T_{n_1}^1}(y_{n_1}^1), G_{T_1^2}(y_1^2) \ldots, G_{T_{n_p}^p}(y_{n_p}^p)) \tag{5}$$

where $C$ denotes the copula function. In our classification problem, it is assume that the observation data are divided into $k$ classes. Therefore, using the previous approaches, we obtain $k$

copulas. The expression of the copula model for class $k$ can be written in following form:

$$H_{T,k}(y_{k,1}^1, \ldots, y_{k,n_p}^p) = C(G_{T_1^1,k}(y_{k,1}^1), \ldots, G_{T_{n_1}^1,k}(y_{k,n_1}^1), G_{T_1^2,k}(y_{k,1}^2) \ldots, G_{T_{n_p}^p,k}(y_{k,n_p}^p)) \tag{6}$$

Let the random variables $(U_{k,1}^1, \ldots, U_{k,n_1}^1, U_{k,1}^2, \ldots, U_{k,n_p}^p)$ be defined as transformations $U_{k,t}^j = G_{T_t^j,k}(y_{k,t}^j)$ that following a uniform distribution on the interval [0,1]. The statistical relationship among these random variables can be expressed through a copula function in the following manner:

$$H(y_{k,1}^1, y_{k,2}^1, \ldots, y_{k,n_p}^p) = C(u_{k,1}^1, u_{k,2}^1, \ldots, u_{k,n_p}^p) \tag{7}$$

Furthermore, the relationship between the joint distribution function and the joint probability density function is expressed as follows:

$$h\left(y_{k,1}^1, y_{k,2}^1, \ldots, y_{k,n_p}^p\right) = \frac{\partial}{\partial y_{k,1}^1 \partial y_{k,2}^1 \ldots \partial y_{k,n_p}^p} H\left(y_{k,1}^1, y_{k,2}^1, \ldots, y_{k,n_p}^p\right)$$

$$= \frac{\partial}{\partial u_{k,1}^1 \partial u_{k,2}^1 \ldots \partial u_{k,n_p}^p} C\left(u_{k,1}^1, u_{k,2}^1, \ldots, u_{k,n_p}^p\right) \prod_{j=1}^p \prod_{t=1}^{n_j} \frac{\partial G_{T_t^j,k}(y_{k,t}^j)}{\partial y_{k,t}^j}$$

$$= c(u_{k,1}^1, u_{k,2}^1, \ldots, u_{k,n_p}^p) \prod_{j=1}^p \prod_{t=1}^{n_j} g_{T_t^j,k}(y_{k,t}^j) \tag{8}$$

With $h(y_{k,1}^1, y_{k,2}^1, \ldots, y_{k,n_p}^p)$ denotes the joint probability density function of the random variables $y_{k,1}^1, y_{k,2}^1, \ldots, y_{k,n_p}^p$. Subsequently, $c(u_{k,1}^1, u_{k,2}^1, \ldots, u_{k,n_p}^p)$ represents the copula function that establishes the connection between the distribution of standardized random variables $u_{k,1}^1, u_{k,2}^1, \ldots, u_{k,n_p}^p$ and $h(y_{k,1}^1, y_{k,2}^1, \ldots, y_{k,n_p}^p)$. Additionally, $g_{T_t^j,k}(y_{k,t}^j)$ denotes the marginal distribution of each respective random variable.

This study employs the Gaussian copula model to characterize the relationships among random variables within images, specifically those associated with partitions and differentiating points. The Gaussian copula function utilized in this research is defined as follows:

$$C(u_{k,1}^1, u_{k,2}^1, \ldots, u_{k,n_p}^p|\mathbf{\Lambda}) = \Phi_{\mathbf{\Lambda}}(\Phi^{-1}(u_{k,1}^1), \Phi^{-1}(u_{k,2}^1), \ldots, \Phi^{-1}(u_{k,n_p}^p)) \tag{9}$$

with $\Phi_{\mathbf{\Lambda}}$ represents the multivariate standard normal distribution function with a mean vector of $\mathbf{0}$ and covariance matrix $\mathbf{\Lambda}$. Additionally, $\Phi^{-1}$ denotes the inverse of the standard normal distribution.

The density function of the Gaussian copula can be derived from the Gaussian copula distribution function as follows:

$$c\left(u_{k,1}^1, u_{k,2}^1, \ldots, u_{k,n_p}^p|\mathbf{\Lambda}\right) = \frac{1}{\sqrt{|\mathbf{\Lambda}|}} \exp\left\{-\frac{1}{2}\mathbf{z}^T\left(\mathbf{\Lambda}^{-1} - I_p\right)\mathbf{z}\right\} \tag{10}$$

With $\mathbf{z} = (\Phi^{-1}(u_{k,1}^1), \Phi^{-1}(u_{k,2}^1), \ldots, \Phi^{-1}(u_{k,n_p}^p))^T$ consists of standardized random variables, and $|\mathbf{\Lambda}|$ represents the determinant of the covariance matrix [41]. The joint density function of

the Gaussian copula can be expressed as follows:

$$h(y^1_{k,1}, y^1_{k,2}, \ldots, y^p_{k,n_p}) = c(u^1_{k,1}, u^1_{k,2}, \ldots, u^p_{k,n_p}) \prod_{j=1}^{p} \prod_{t=1}^{n_j} g_{T^j_t,k}(y^j_{k,t})$$

$$= \frac{1}{\sqrt{|\Lambda)}} \exp\left\{-\frac{1}{2} z^T (\Lambda^{-1} - I_p) z\right\} \prod_{j=1}^{p} \prod_{t=1}^{n_j} g_{T^j_t,k}(y^j_{k,t}) \tag{11}$$

The joint density function ($h$) depicts the pattern of relationships among random variables associated with partitions and distinguishing points.

## Parameter estimation

The Inference Function for Marginals (IFM) is frequently utilized in estimating copula model parameters. IFM aims to identify optimal values for the model parameters by maximizing the likelihood function, representing the probability of observing the data under the copula model [42]. This likelihood function is based on the joint probability density function of the data, influenced by the copula parameters. The formulation of the likelihood function is as follows:

$$L(\Theta, \Lambda | y^1_{k,1}, y^1_{k,2}, \ldots, y^p_{k,n_p}) = \prod_{i=1}^{N_k} h(y^1_{k,1}, y^1_{k,2}, \ldots, y^p_{k,n_p})$$

$$= \prod_{i=1}^{N_k} [c(u^1_{k,1}, u^1_{k,2}, \ldots, u^p_{k,n_p}) \prod_{j=1}^{p} \prod_{t=1}^{n_j} g_{T^j_t,k}(y^j_{k,t})] \tag{12}$$

with $\Lambda$ represents the copula parameter, and $\Theta$ is the marginal parameter. In parameter estimation, the IFM often involves the use of the logarithm of the likelihood function [43]. The log-likelihood function for the copula is given as follows:

$$\ell(\Theta, \Lambda | y^1_{k,1}, y^1_{k,2}, \ldots, y^p_{k,n_p}) = \log L(\Theta, \Lambda | y^1_{k,1}, y^1_{k,2}, \ldots, y^p_{k,n_p})$$

$$= \sum_{i=1}^{N_k} \log c(u^1_{k,1}, u^1_{k,2}, \ldots, u^p_{k,n_p}) + \sum_{j=1}^{p} \sum_{t=1}^{n_j} \sum_{i=1}^{N_k} \log g_{T^j_t,k}(y^j_{k,t}) \tag{13}$$

for the Gaussian copula, the likelihood function is given as follows:

$$\ell(\Theta, \Lambda | y^1_{k,1}, y^1_{k,2}, \ldots, y^p_{k,n_p})$$
$$= \ell_c(\hat{\Theta}, \Lambda | y^1_{k,1}, y^1_{k,2}, \ldots, y^p_{k,n_p}) + \sum_{i=1}^{N_k} \ell_{g_{T^j_t,k}}(\theta_{k,j,t} | y^1_{k,1,i}, y^1_{k,2,i}, \ldots, y^p_{k,n_p,i}) \tag{14}$$

with $\Theta = [\theta^1_1, \theta^1_2, \ldots, \theta^p_{n_p}]$.

The IFM method utilizes the marginal likelihood to estimate marginal parameters and the copula likelihood to estimate copula parameters. The approach involves a two-step process: first, estimating the marginal parameters and then using these estimates as inputs for the copula likelihood function to estimate copula parameters [40, 41]. The log-likelihood function of the Gaussian copula can be decomposed into two distinct components: the log-likelihood function component of the copula, given in Eq (15), and the log-likelihood function component of the marginal, given in Eq (16) as follows:

$$\ell_c\left(\hat{\Theta}, \Lambda | y^1_{k,1}, y^1_{k,2}, \ldots, y^p_{k,n_p}\right) = \frac{N_k}{2} \log|\Lambda| - \sum_{i=1}^{N_k} -\frac{1}{2} z^T_i (\Lambda^{-1} - I) z_i \tag{15}$$

$$\sum_{i=1}^{N_k} \ell_{g_{T^j_t,k}}(\theta_{k,j,t} | y^1_{k,1,i}, y^1_{k,2,i}, \ldots, y^p_{k,n_p,i}) = \sum_{j=1}^{p} \sum_{t=1}^{n_j} \sum_{i=1}^{N_k} \log g_{T^j_t,k}(y^j_{k,t}) \tag{16}$$

The optimization process is initiated by addressing the marginal components to derive the parameters of the marginal distribution [44–46]. The IFM algorithm is structured as follows:

**Step 1**: Determine the log-likelihood function for the Gaussian copula using Eq (14)

**Step 2**: Obtain the marginal parameter estimates by maximizing the log-likelihood function for the marginal, as given by Eq (16). For partitions $j = 1,2,\ldots,p$, and points $t = 1,2,\ldots,n_j$:

$$\hat{\theta}_{g_{T_t^j,k}} = \text{Argmax}\ell_{g_{T_t^j,k}} \tag{17}$$

**Step 3**: Substitute $\hat{\theta}_{g_{T_t^j,k}}$ from step 2 to obtain the log-likelihood function for the Gaussian copula using Eq (15).

**Step 4**: Obtain the Gaussian copula parameter estimates by maximizing the log-likelihood function for Gaussian copula as follows:

$$\hat{\Lambda} = \underset{\rho}{\text{argmax}}\,\ell_c(\hat{\Theta}, \Lambda | y_{k,1}^1, y_{k,2}^1, \ldots, y_{k,n_p}^p) \tag{18}$$

Therefore, the parameter estimates using the IFM method are acquired by satisfying the following equations:

$$\left(\frac{\partial}{\partial\theta_{k,1}^1}\ell_{g_{T_1^1,k}}, \frac{\partial}{\partial\theta_{k,2}^1}\ell_{g_{T_2^1,k}}, \ldots, \frac{\partial}{\partial\theta_{k,n_p}^p}\ell_{g_{T_{n_p}^p,k}}, \frac{\partial}{\partial\rho}\ell_c\right) = (0, \ldots, 0) \tag{19}$$

The procedure for estimating parameters in the marginal distribution is contingent upon the specific nature of the chosen distribution. Within the Gaussian copula model framework, the marginal distribution may be parametric or non-parametric. An exemplification of a parametric distribution is the beta distribution, defined by a domain function with values confined to the interval [0, 1]. The probability density function of the beta distribution for each $j = 1,2,\ldots,p$, and $t = 1,2,\ldots,n_j$ is expressed as follows:

$$g_{T_t^j,k}\left(y_{k,t}^j | \theta_{1,T_t^j,k}, \theta_{2,T_t^j,k}\right) = \frac{1}{B(\theta_{1,T_t^j,k}, \theta_{2,T_t^j,k})} y_{k,t}^j \theta_{1,T_t^j,k} - 1(1 - y_{k,t}^j)^{\theta_{2,T_t^j,k}} \tag{20}$$

Where $0 \leq y_{k,t}^j \leq 1$ and $\theta_{1,T_t^j,k}, \theta_{2,T_t^j,k} > 0$, with $\theta_{1,T_t^j,k}$ and $\theta_{2,T_t^j,k}$ denoting the parameters of Beta distribution. These parameters are estimated through the Maximum Likelihood Estimation (MLE) method, utilizing the log-likelihood Beta function as follows:

$$\ell_{g_{T_t^j,k}}(\theta_{k,t}^j | y_{k,1,i}^1, y_{k,2,i}^1, \ldots, y_{k,n_p,i}^p) = \log\left(\sum_{i=1}^{N_k} g_{T_t^j,k}(y_{k,t}^j | \theta_{1,T_t^j,k}, \theta_{2,T_t^j,k})\right) \tag{21}$$

## Classification and evaluation

In this study, the employed classification method adopts a probabilistic framework [29], articulated as follows:

$$P(C_k | \dot{X}) = \frac{P(C_k, \dot{X})}{P(\dot{X})} \quad \text{with } k = 0, 1, \ldots, (v-1) \tag{22}$$

Here, $\dot{X}$ represents the new input (testing), and $C_k$ denotes the class of image $k$. Consequently, $P(C_k | \tilde{X})$ signifies the conditional probability of class $k$ given the input.

$$P(C_k, \dot{X}) = P(\dot{X} | C_k)P(C_k) \tag{23}$$

The term $P(\dot{X}|C_k)$ embodies the joint distribution modeled by the Gaussian copula. Additionally, $P(C_k)$ denotes the proportion of training data for class $k$, computed as follows:

$$P(C_k) = \frac{N_k \ training}{N \ training} \tag{24}$$

Where $N_k \ training$ represents the number of training samples for class $k$, and $N \ training$ signifies the total quantity of training data. Consequently, the conditional probability $P(C_k|\dot{X})$ is expressed as:

$$P(C_k|\dot{X}) = \frac{P(\dot{X}|C_k)P(C_k)}{P(\dot{X})} \tag{25}$$

After completing the classification process, the next step is model evaluation. Model evaluation ensures the model's efficacy in predicting the image class. In this study, model evaluation is performed using the confusion matrix. Classification performance, in this context, is assessed using accuracy metrics. In addition, precision, recall, and F1 scores are used to comprehensively assess the model's performance. Similar studies also use other metrics for classification evaluation [47]. These evaluation metrics measure image classification's accuracy, completeness, and effectiveness.

The analysis steps of this study can be illustrated in a research flowchart that reflects the main steps. Initially, a pre-processing phase is carried out, including data cleaning, normalization, and splitting the data into training and testing sets. Next, the data preparation for training and testing is conducted, followed by the formation of ECDF and DFDV as random variables in the Gaussian copula model. Then, the formulation of the Gaussian copula model is executed to capture the dependencies among random variables, involving the estimation of model parameters. This estimation is obtained by applying the IFM method to achieve optimal estimates. The subsequent step involves the classification process using the developed model, followed by the evaluation and testing phase using specific evaluation metrics. It is important to note that the parameters used in the evaluation process of the testing data are the parameters obtained from the training model. The research flowchart is presented in Fig 1.

## Data characterization and pre-processing

This study introduces a Gaussian copula model for image classification. The data used in this study is MNIST handwritten numeric image data. The dataset consists of 60,000 grayscale images depicting numbers 0–9 for training and an additional 10,000 for testing. As part of the preprocessing step, the data undergoes normalization, ensuring each image is represented as a $20 \times 20$-pixel array to effectively capture the edges of handwritten numerals. Normalization ensures uniformity in the data, allowing the model to focus on the essential features of the numerals without being affected by variations in size or intensity. After normalization, the partitioning process was run, resulting in systematically organized partitions that included complete, two-line, three-line, and four-line data sets. This partitioning enhances the model's ability to discern patterns and features within the handwritten numerical images visually depicted in Fig 2 as follows:

This study uses the full data and various partitions to investigate the effect of the number of partitions on the performance of the classification model. A larger number of partitions may lead to a scenario where there are too few pixels in each partition, thus prompting the restriction to four rows. Partitioning the data aims to analyze the impact of different levels of detail on the model's accuracy. A key consideration involves striking a balance between achieving a

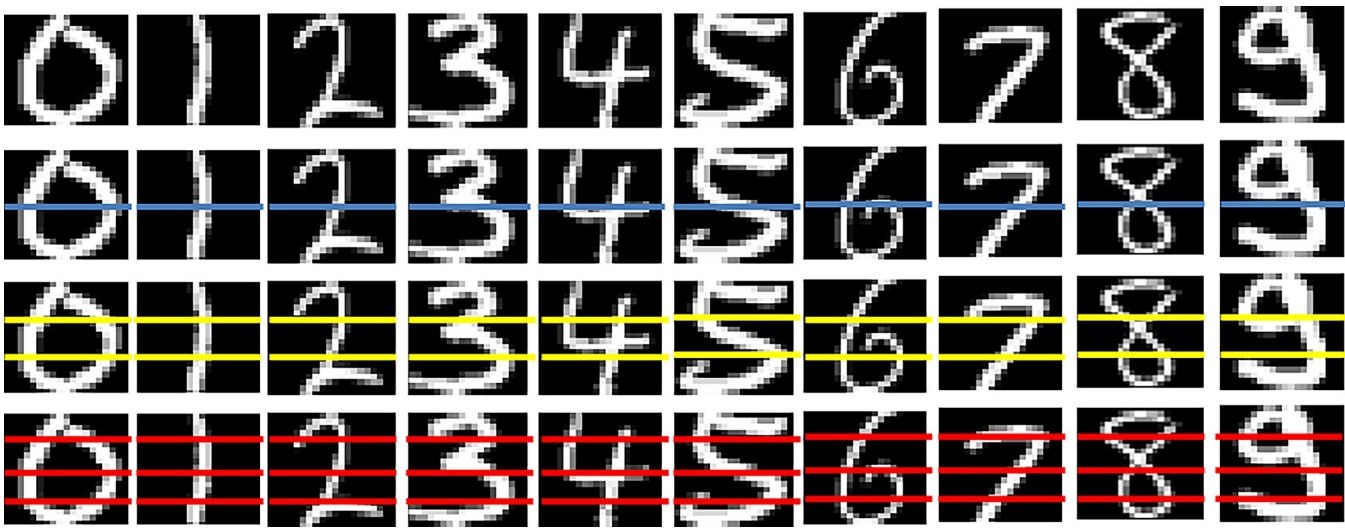

**Fig 1. Block diagram.** Illustrating stages and processes in image classification using Gaussian copula with symbolic data ECDF and DFDV.

**Fig 2. MNIST data image partitions: Full, two-row, three-row, and four-row partitions.**

more detailed image representation through partitioning and ensuring an adequate number of pixels for stable analysis. Too few pixels in a partition could lead to a loss of critical information, while too many partitions could introduce noise and complicate the classification task.

The process involves representing image features through a matrix of pixel intensities. To streamline data representation and analysis, the matrix of pixel values transforms into a vector, a procedure commonly known as vectorization. Adopting a row-by-row approach, each row of the pixel value matrix is systematically extracted and merged into a data vector, as outlined [48]. Consequently, each pixel value matrix is transformed into a corresponding data vector. This transformation facilitates the handling and processing of image data by converting it into a more manageable form for machine learning algorithms. When row-wise vectorization is applied to a matrix denoted as $A_{20 \times 20}$.

The elements of the data matrix A have components $x$, which is a representation of the pixel intensity values. We convert each x element of the data matrix, initially in the range [0,255], to the range [0,1] by dividing it by 255, expressed as $\tilde{x} = x/255$. This transformation results in $\tilde{x}$ values that are easy to handle in numerical calculations, especially in machine learning algorithms and statistical analyses, which are more stable and fast when performed in the interval [0,1]. When the vectorization process converts the data matrix A into a vector, the result is the vector Vec(A). Moreover, the transformation of matrix A yields the matrix $\tilde{A}$, which subsequently transforms into the vector Vec($\tilde{A}$). Next, the ECDF is constructed by following the method specified in Eq (1). Fig 3 visually illustrates the construction of the ECDF. The ECDF serves as an image characteristic, capturing the pixel values that are less than or equal to x. These characteristics show variations in different image classes, underscoring the importance of identifying specific realization points of $x$ that can effectively distinguish them. The ECDF [rovides a statistical summary of the image data, highlighting key features that contribute to the classification performance. The resulting vector Vec($\tilde{A}$)$_{1 \times 400}$ and ECDF $\mathcal{F}(\tilde{x})$ is visually illustrated in Fig 3:

## ECDF and DFDV results

Each image generates an ECDF with a total of N images, and a set of ECDF is produced for these images. At specific points, denoted as $T_1$ and $T_2$, the ECDF values from each function are gathered to construct a Distribution Function of Distribution Values (DFDV) as defined in Eq (2). This DFDV represents the distribution function of pixel ECDF values, illustrating the characteristics of pixel value utilization at points $T_1$ and $T_2$. These characteristics can vary across image classes, and selecting different points will result in different characteristics, thus referred to as differentiating points. In this case the specific distinguishing points selected are $T_1 = 0.1$ and $T_2 = 0.7$. The results of ECDF formation for each class in the full dataset are presented in Fig 4:

The displayed image shows an ECDF plot of the value $\tilde{x}$ for digits 0 to 9 (k = 0 to k = 9) in the MNIST data set. Each subplot describes the cumulative distribution of normalized pixel values $\tilde{x}$ to certain digits, from 0 to 9. Each subplot's X axis displays a normalized pixel value, meaning it divides the original pixel value by 255 to fall within the range [0,1]. The $Y$ axis indicates the cumulative value of the ECDF, which reflects the accumulative proportion of pixels with an intensity less than or equal to the value on the X axis. The red and blue vertical lines mark the specific distinguishing points selected are $T_1 = 0.1$ and $T_2 = 0.7$

The ECDF plot illustrates the distribution of normalized pixel intensity values for each digit in the MNIST data set. By looking at the variations and curve shapes in each subplot, we can obtain information about the distribution patterns of unique pixel strength values for every

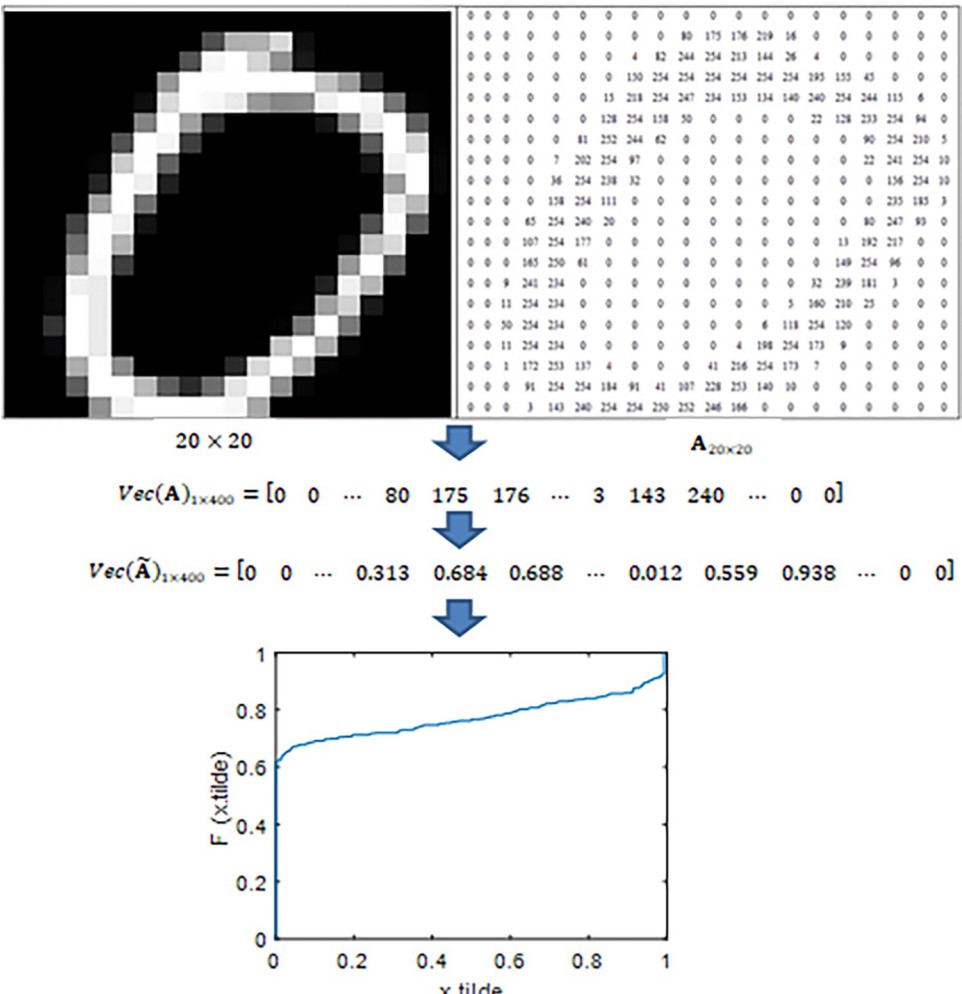

**Fig 3. Illustration of ECDF formation: Images represented in pixel intensity matrix, vectorized, transformed, and ECDF $F(\tilde{x})$ formed using Eq (1).** Here, x.tilde represent $\tilde{x}$, and F(x.tilde) represent $\mathcal{F}(\tilde{x})$.

digit. This analysis is useful to understand how the pixel value distribution features of different digit images behave and can help in the development of accurate classification models.

The results of the ECDF analysis reveal distinctive patterns among classes, indicating variations in the characteristics of pixel intensity distributions across different classes. A notable disparity is observed in the ECDF for the digits 0 and 1. The distribution of the ECDF values at the differential points effectively represents the visible differences in distribution characteristics between these image classes. Significant differences in ECDF digits 0 and 1 indicate that the intensity of pixels and their distribution patterns are different, which can help distinguish between different digits. For example, a 0 digit may have a more uniform pixel distribution compared to a 1 digit, which tends to have greater intensity variations in some areas of the image. These discernible differences in distribution characteristics among image classes can be effectively represented by the distribution of ECDF values at differentiating points.

A DFDV is constructed at each differentiating point from the ECDF values set. This DFDV serves as the image characteristics for each partition and differentiating point. This DFDV effectively describes the probability distribution of image features with values less than or equal to a certain value. This DFDV is interconnected as it is derived from the same set of

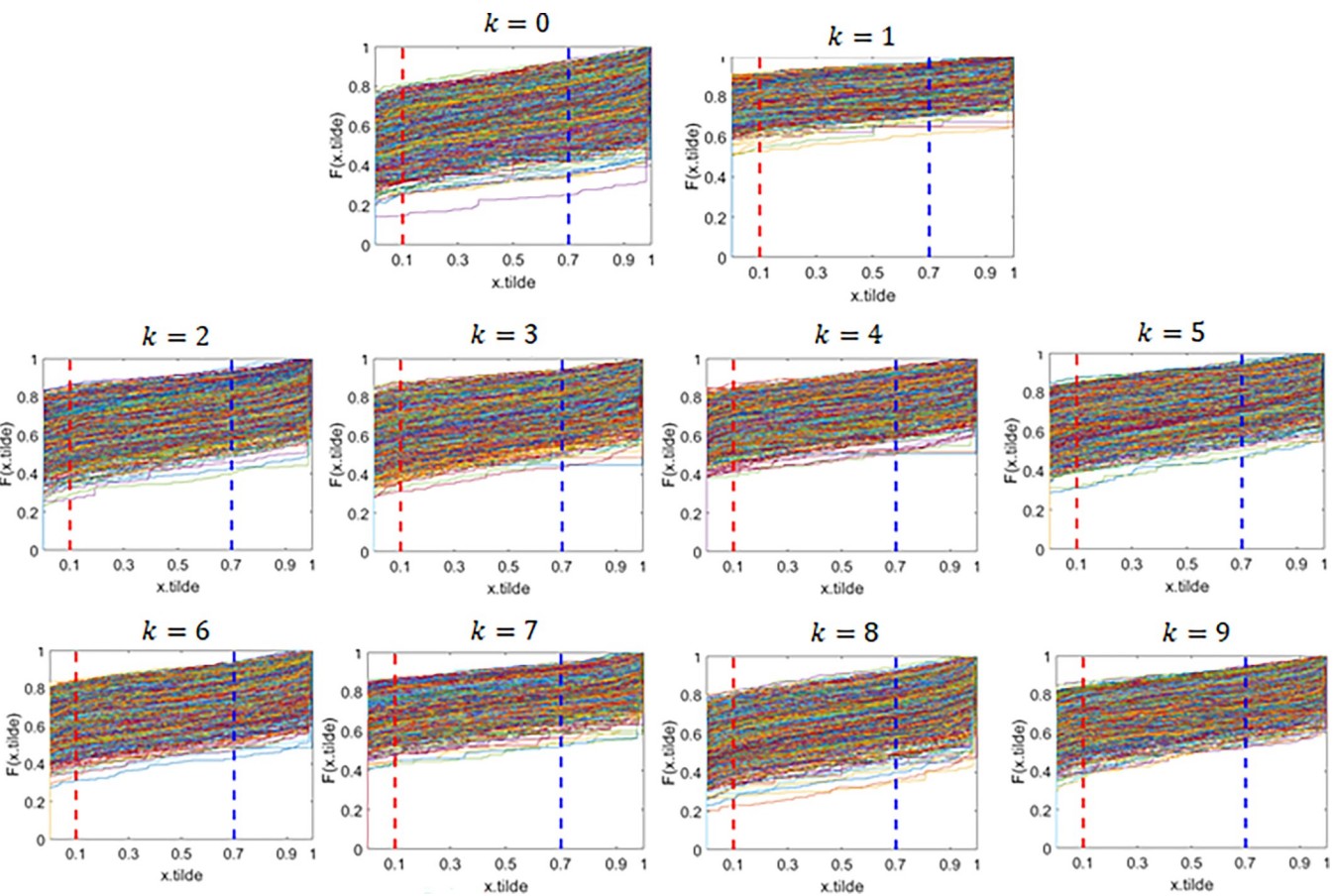

**Fig 4. ECDF patterns across classes with full dataset.** Here, x.tilde represent $\tilde{x}$, and F(x.tilde) represent $\mathcal{F}(\tilde{x})$.

images. This interdependence is modeled in terms of Gaussian copulas, with the DFDV treated as random variables. The DFDV formation process and Gaussian copulas are visually depicted in Fig 5:

The interdependence of distribution values within a single image at differentiating points is modelled in this study using a copula, with $G_{T_1}(y)$ and $G_{T_2}(y)$ serving as random variables. Various copula models, including Gaussian, Clayton, Frank, t-Student, and Gumbel, are available for consideration. The choice of the copula type is determined based on the scatter plot of the ECDF values at points $T_1$ and $T_2$ as random variables. The patterns observed in the scatter diagram are utilized to identify the most suitable copula model [48]. This approach employs graphical representations to discern the optimal copula model based on the observed relationships between random variables. The scatter plot results for $G_{T_1}(y)$ and $G_{T_2}(y)$ for classes $k = 0,1,\ldots,9$ are presented in Fig 6:

After selecting the Gaussian copula model, the next step is to specify the DFDV at points $T_1$ and $T_2$. These DFDV values capture the distribution characteristics at these points and are crucial for further analysis within the Gaussian copula framework. The DFDV results for classes $k = 0,1,\ldots,9$ at $T_1 = 0.1$ and $T_2 = 0.7$ are shown in Fig 7.

The DFDV results for the image classes at the distinguishing points show different distributions. Visually, the separate shapes of these distributions indicate that the characteristics of the image classes are different. This highlights the ability of DFDV to capture and represent the

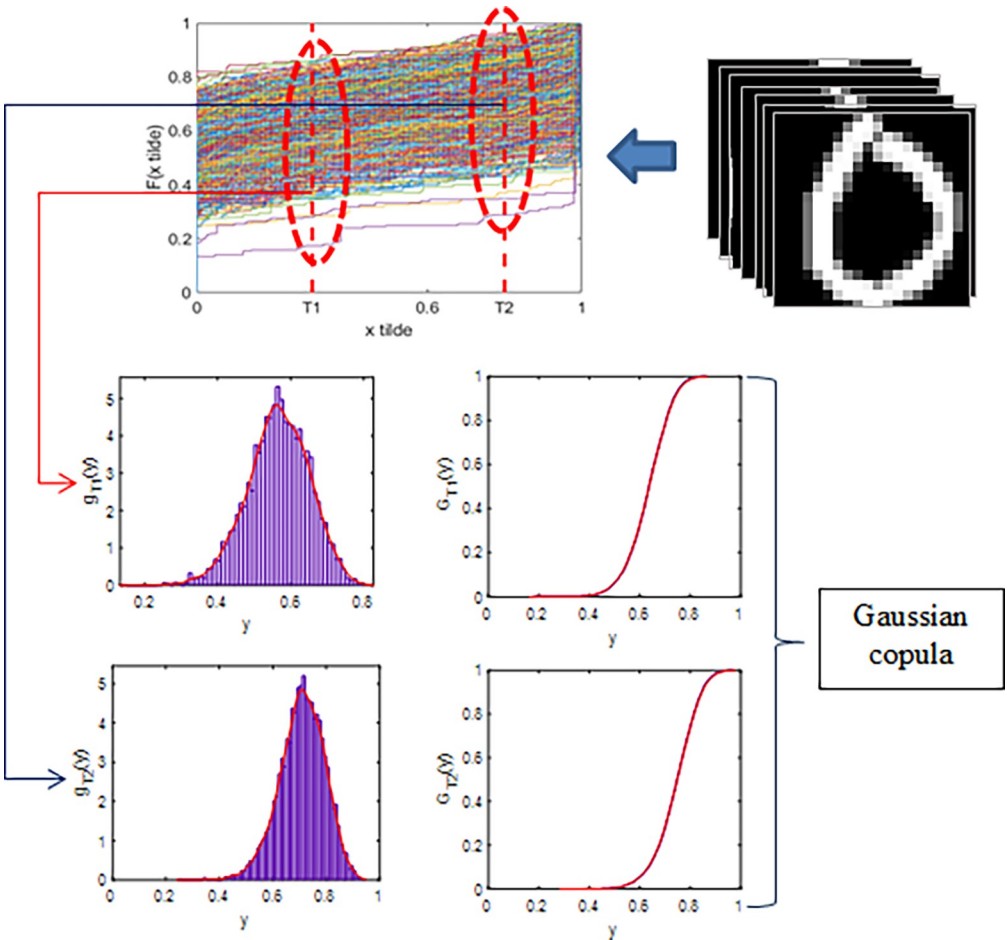

**Fig 5. Illustration of DFDV formation at differentiating points for full dataset.** Formed DFDV Used as Random Variables in Gaussian Copula Model. Here, x.tilde represent $\tilde{x}$, and F(x.tilde) represent $\mathcal{F}(\tilde{x})$.

unique features of each class, which is crucial for accurate classification and further analysis. By providing a clear differentiation between the classes, DFDV enhances the model's ability to accurately identify and classify images based on their distinctive features.

## Gaussian copula results

The Gaussian copula model applied to image classes $k = 0,1,\ldots,9$, for the full dataset, specifically with $j = 1$ (only one partition) and differentiating points at $t = 1,2$, as described in Eq (8), is formulated as follows:

$$C(u_{k,1}^1, u_{k,2}^1 | \Lambda) = \Phi_\Lambda(\Phi^{-1}(u_{k,1}^1), \Phi^{-1}(u_{k,2}^1)) \tag{26}$$

The term $u_{k,t}^j$ represents the random variable $U_{k,t}^j = G_{T_t^j,k}(y_{k,t}^j)$, which follows a uniform distribution on the interval [0,1]. The notation $\Lambda$ denotes the covariance matrix serving as parameter for Gaussian copula. The function $\Phi_\Lambda$ represents the multivariate standard normal distribution, and $\Phi^{-1}$ denotes the inverse of the standard normal distribution. This model provides a mathematical representation that captures the dependency structure among various image classes, emphasizing the relationships between variables within the dataset.

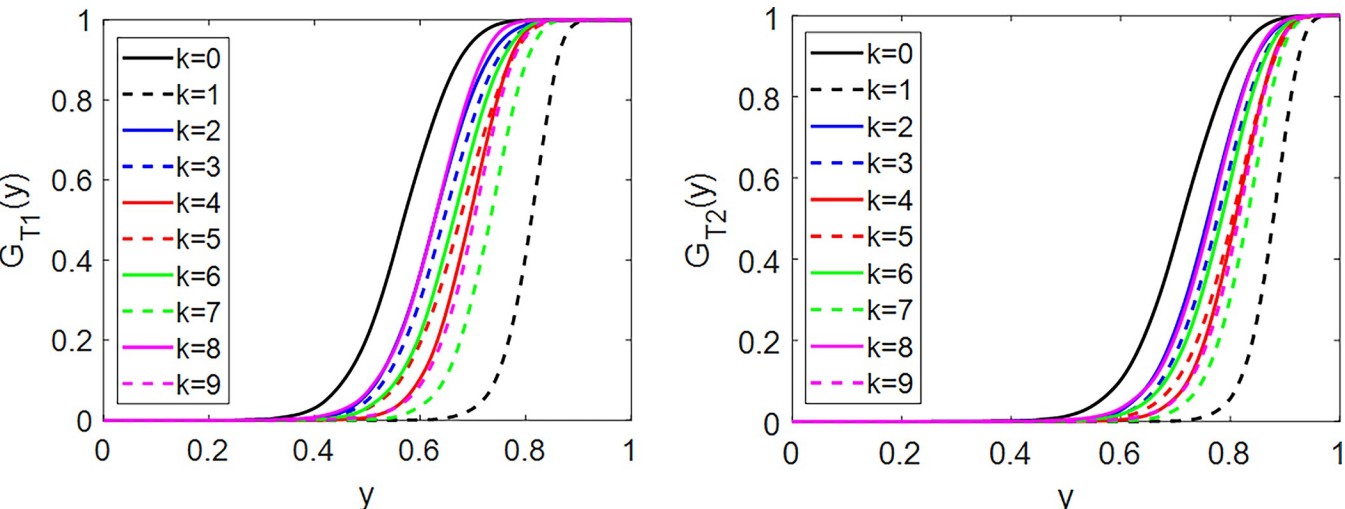

**Fig 6. Scatter plot analysis of $G_{T_1}(y)$ and $G_{T_2}(y)$.** Indicates Gaussian Copula Model with Linear and Elliptical Patterns. Here, x.tilde represent $\tilde{\tilde{x}}$.

**Fig 7. DFDV $G_{T_1}(y)$ and $G_{T_2}(y)$ at $T_1 = 0.1$ and $T_2 = 0.7$ for image class $k = 0, 1, \ldots, 9$.** Distinct distributions are evident among different classes.

The model presented in Eq (26) involves two random variables, and the density model of the Gaussian copula can be expressed as follows:

$$c\left(u_{k,1}^1, u_{k,2}^1 | \boldsymbol{\Lambda}\right) = \frac{1}{\sqrt{|\boldsymbol{\Lambda}|}} \exp\left\{-\frac{1}{2}\boldsymbol{z}^T\left(\boldsymbol{\Lambda}^{-1} - \boldsymbol{I}_2\right)\boldsymbol{z}\right\} \tag{27}$$

with $\boldsymbol{z} = (\Phi^{-1}(u_{k,1}^1)\Phi^{-1}(u_{k,2}^1))^T$. This function characterizes the dependency structure between two specific random variables, influenced by the parameter matrix $\boldsymbol{\Lambda}$. The Gaussian copula joint density function of Eq (27) can be given as follows:

$$h(y_{k,1}^1, y_{k,2}^1) = c(u_{k,1}^1, u_{k,2}^1) \prod_{t=1}^2 g_{T_t^1,k}(y_{k,t}^1)$$

$$= \frac{1}{\sqrt{|\Lambda|}} \exp\left\{-\frac{1}{2}\boldsymbol{z}^T(\Lambda^{-1} - I_2)\boldsymbol{z}\right\} \prod_{t=1}^2 g_{T_t^1,k}(y_{k,t}^1) \tag{28}$$

The probability density function described here comprises two components. The initial component encapsulates the interdependence structure among the random variables within the copula function, while the subsequent component denotes the marginal distribution for each random variable. The copula function serves to characterize the collective behavior of the random variables, capturing their dependence structure. Parameter estimation for this model is undertaken through the Inference Functions for Margins (IFM) method. This approach utilizes the log-likelihood function, as delineated in Eq (13).

$$\ell(\boldsymbol{\Theta}, \boldsymbol{\Lambda}|y_{k,1}^1, y_{k,2}^1) = \sum_{i=1}^{N_k} \log c(u_{k,1}^1, u_{k,2}^1) + \sum_{i=1}^{N_k} \sum_{t=1}^2 \log g_{T_t^1,k}(y_{k,t,i}^1)$$

$$= l_c(\hat{\boldsymbol{\Theta}}, \boldsymbol{\Lambda}|\boldsymbol{y}_{k,1}^1, \boldsymbol{y}_{k,2}^1) + \sum_{i=1}^{N_k} l_{g_{T_t^1,k}}(\boldsymbol{\theta}_{k,t}^1|\boldsymbol{y}_{k,1,i}^1, \boldsymbol{y}_{k,2,i}^1)$$

with $\boldsymbol{\Theta} = [\theta_{k,1}^1, \theta_{k,2}^1]$, and

$$\ell_c\left(\hat{\boldsymbol{\Theta}}, \boldsymbol{\Lambda}|y_{k,1}^1, y_{k,2}^1\right) = \frac{N_k}{2}\log|\boldsymbol{\Lambda}| - \sum_{i=1}^{N_k} -\frac{1}{2}\boldsymbol{z}_i^T\left(\boldsymbol{\Lambda}^{-1} - I_2\right)\boldsymbol{z}_i$$

$$\sum_{i=1}^{N_k} \ell_{g_{T_t^1,k}}(\theta_{g_{T_t^1,k}}|y_{k,1,i}^1, y_{k,2,i}^1) = \sum_{i=1}^{N_k} \sum_{t=1}^2 \log g_{T_t^1,k}(y_{k,t,i}^1)$$

The parameter estimation process starts with the marginal component, which includes the distribution of each random variable. After obtaining the parameters for the marginal distribution, parameter estimation for the Gaussian copula component follows. In this study, the marginal distribution chosen is the beta distribution since the domain of the random variable is in the interval [0,1]. The results of the beta parameter estimation using Eq (16) are presented as Table 1 below:

Table 1 shows the beta parameter estimates for the Gaussian copula model for various image classes. Comparison between classes and observations at points $T_1 = 0.1$ and $T_2 = 0.7$ provides insight into the variability of the dependence and sensitivity of the model at those points. The beta parameter estimates provide information on the dependence structure and characteristics of the pixel intensity distribution.

Parameter estimation is conducted by maximizing the Gaussian copula log-likelihood function using the IFM method, as expressed in Eq (17). The estimation results of the covariance matrix parameters are obtained as Table 2 follows:

**Table 1. Beta distribution parameter estimation results.** Case $k = 0,1,\ldots,9$, $j = 1$ and $t = 1,2$ for random variables $u_{k,1}^1$ and $u_{k,2}^1$.

| Image class | Beta distribution parameters | | | |
| :---: | :---: | :---: | :---: | :---: |
| | $T_1 = 0.1$ | | $T_2 = 0.7$ | |
| | $\theta_1$ | $\theta_2$ | $\theta_1$ | $\theta_2$ |
| 0 | 20.570 | 12.696 | 20.148 | 8.306 |
| 1 | 59.749 | 12.183 | 53.121 | 7.662 |
| 2 | 24.183 | 11.785 | 24.734 | 8.049 |
| 3 | 23.777 | 10.745 | 25.380 | 7.545 |
| 4 | 35.526 | 13.130 | 33.223 | 8.118 |
| 5 | 24.891 | 9.879 | 26.333 | 6.744 |
| 6 | 26.527 | 11.279 | 26.284 | 7.549 |
| 7 | 40.104 | 12.533 | 38.265 | 8.170 |
| 8 | 25.713 | 12.683 | 24.792 | 8.007 |
| 9 | 35.113 | 12.624 | 33.616 | 8.011 |

The covariance matrix values appear relatively high, indicating a strong relationship between the random variables $u_{k,1}^1$ and $u_{k,2}^1$. However, these values show minimal variation across classes, indicating that, with the full dataset, the characteristics that distinguish between classes cannot be adequately discerned. The introduction of image partitioning techniques becomes very important to improve the discrimination of object characteristics among image classes.

In the case of partitioning, the formation of ECDF and DFDV is performed for each partition. The random variables used in the Gaussian copula model depend on the number of partitions and discrimination points defined. For example, with two discrimination points, the complete data set model consists of two random variables. When the partitions are two, four random variables are involved, increasing to six for three partitions and eight for four partitions. Using the Gaussian copula model, as articulated in Eq (9) and the joint probability density model described in Eq (10), the Gaussian copula model is formulated for each scenario as Table 3 follows:

**Table 2. Covariance matrix parameter estimation results.** Case $k = 0,1,\ldots,9$, $j = 1$ and $t = 1,2$ for random variables $u_{k,1}^1$ and $u_{k,2}^1$.

| $k$ | Covariance matrix ($\Lambda$) | $k$ | Covariance matrix ($\Lambda$) |
| :---: | :---: | :---: | :---: |
| 0 | $\begin{bmatrix} 1 & 0,968 \\ 0,968 & 1 \end{bmatrix}$ | 5 | $\begin{bmatrix} 1 & 0,949 \\ 0,949 & 1 \end{bmatrix}$ |
| 1 | $\begin{bmatrix} 1 & 0,942 \\ 0,942 & 1 \end{bmatrix}$ | 6 | $\begin{bmatrix} 1 & 0,947 \\ 0,947 & 1 \end{bmatrix}$ |
| 2 | $\begin{bmatrix} 1 & 0,947 \\ 0,947 & 1 \end{bmatrix}$ | 7 | $\begin{bmatrix} 1 & 0,950 \\ 0,950 & 1 \end{bmatrix}$ |
| 3 | $\begin{bmatrix} 1 & 0,951 \\ 0,951 & 1 \end{bmatrix}$ | 8 | $\begin{bmatrix} 1 & 0,946 \\ 0,946 & 1 \end{bmatrix}$ |
| 4 | $\begin{bmatrix} 1 & 0,951 \\ 0,951 & 1 \end{bmatrix}$ | 9 | $\begin{bmatrix} 1 & 0,944 \\ 0,944 & 1 \end{bmatrix}$ |

**Table 3. Gaussian copula models for $j$ = 1,2,3,4 cases.** These Models are Applied to Image Class $k$ = 0,1,...,9.

| Partitions (rows) | Gaussian copula model |
| --- | --- |
| Full data | $h\left(y_{k,1}^{1}, y_{k,2}^{1}\right) = \frac{1}{\sqrt{\lvert\Lambda\rvert}}\exp\left\{-\frac{1}{2}\mathbf{z}^{T}\left(\Lambda^{-1} - I_{2}\right)\mathbf{z}\right\}\prod_{t=1}^{2} g_{T_{t}^{1},k}(y_{k,t}^{1})$ |
| 2 | $h\left(y_{k,1}^{1}, \ldots, y_{k,2}^{2}\right) = \frac{1}{\sqrt{\lvert\Lambda\rvert}}\exp\left\{-\frac{1}{2}\mathbf{z}^{T}\left(\Lambda^{-1} - I_{4}\right)\mathbf{z}\right\}\prod_{j=1}^{2}\prod_{t=1}^{2} g_{T_{t}^{j},k}(y_{k,t}^{j})$ |
| 3 | $h\left(y_{k,1}^{1}, \ldots, y_{k,2}^{3}\right) = \frac{1}{\sqrt{\lvert\Lambda\rvert}}\exp\left\{-\frac{1}{2}\mathbf{z}^{T}\left(\Lambda^{-1} - I_{6}\right)\mathbf{z}\right\}\prod_{j=1}^{3}\prod_{t=1}^{2} g_{T_{t}^{j},k}(y_{k,t}^{j})$ |
| 4 | $h\left(y_{k,1}^{1}, \ldots, y_{k,2}^{4}\right) = \frac{1}{\sqrt{\lvert\Lambda\rvert}}\exp\left\{-\frac{1}{2}\mathbf{z}^{T}\left(\Lambda^{-1} - I_{8}\right)\mathbf{z}\right\}\prod_{j=1}^{4}\prod_{t=1}^{2} g_{T_{t}^{j},k}(y_{k,t}^{j})$ |

## Classification and evaluation results

After obtaining the Gaussian copula model, characterized by the joint density function with marginal distribution parameters from Table 2 and Gaussian copula parameters from Table 3, the subsequent phase involves the classification process. The classification is executed through the probabilistic method outlined in Eqs (21–23). Before commencing the classification stage, the initial step involves preparing the training data, which includes pre-processing, image pixel extraction, data transformation, and ECDF formation. ECDF values are determined at the same distinguishing points utilized in the training process. The outcomes of ECDF testing at discrimination points $T_1$ = 0.1 and $T_2$ = 0.7 are presented as Fig 8 below:

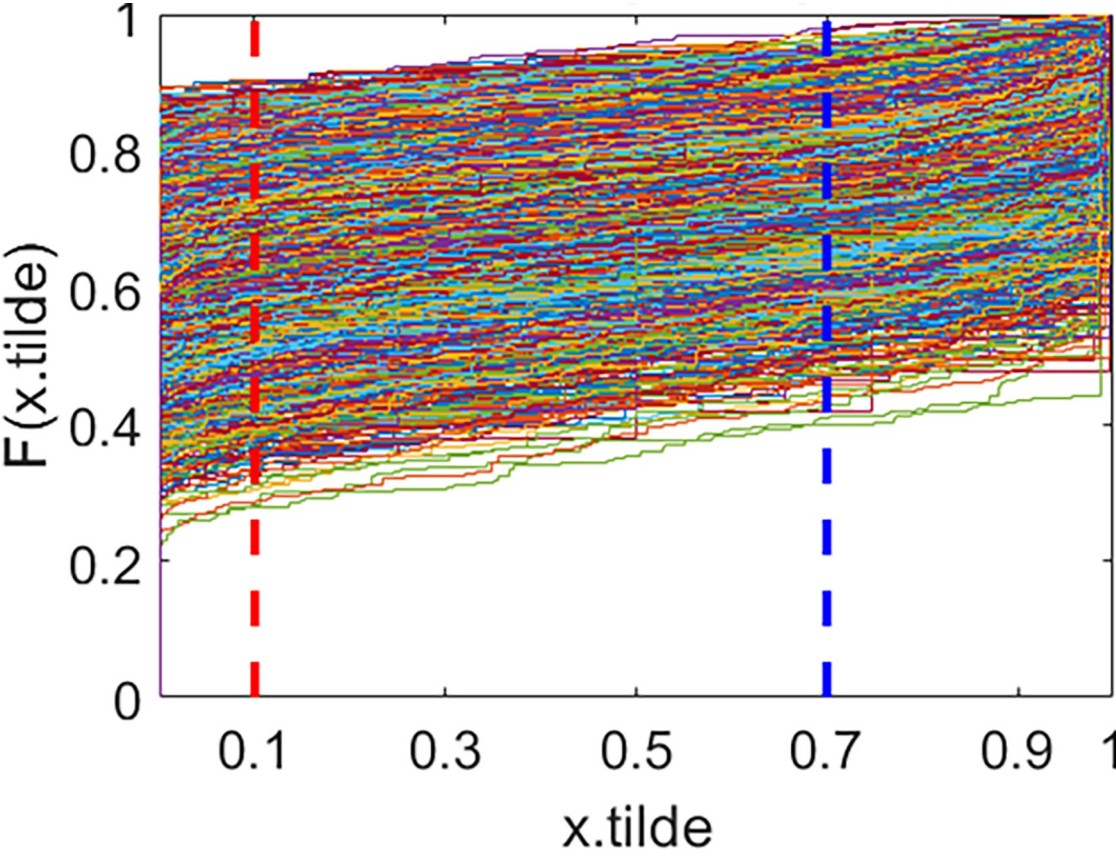

**Fig 8. ECDF results for testing data.** Case $j$ = 1 and $T_1$ = 0.1 and $T_2$ = 0.7.

**Table 4. The results of evaluation metrics values (%).** Case $j = 1,2,3,4$ and $T_1 = 0.1$ and $T_2 = 0.7$.

| Partitions | Average | | | | Maximum | | | |
|---|---|---|---|---|---|---|---|---|
| | Accuracy | Precision | Recall | F1-score | Accuracy | Precision | Recall | F1-score |
| 1 (full data) | 29.47 | 20.70 | 26.40 | 23.20 | 84.30 | 81.80 | 84.50 | 83.13 |
| 2 | 48.14 | 55.70 | 47.47 | 51.25 | 94.36 | 90.90 | 91.60 | 91.25 |
| 3 | 58.73 | 67.20 | 54.30 | 60.07 | 94.80 | 94.20 | 93.90 | 94.05 |
| 4 | 62.22 | 65.00 | 62.46 | 63.70 | 96.92 | 95.30 | 95.30 | 95.30 |

Next is the classification process using the input ECDF values at points $T_1 = 0.1$ and $T_2 = 0.7$ for the training data. The classification results are given as Table 4 follows:

Table 4 presents the results of classification evaluation using the Gaussian copula model in various data partition scenarios. This evaluation includes several classification performance metrics, such as Accuracy, Precision, Recall, and F1-score. These results reflect the model's performance when using the entire dataset without partitioning. The low accuracy indicates that the model needs help to classify diverse classes.

Dividing the data into two partitions significantly improves accuracy and other performance metrics. The model performs better in recognizing and separating features in both partitions. This improvement continues as the data is divided into three partitions, with accuracy and other performance metrics improving further. This suggests that using three partitions provides a better representation of the dataset's variation.

With data divided into four partitions, there is a significant improvement in all performance metrics. The model achieves its highest performance with four partitions, indicating that more partitions provide a better representation and support more detailed feature separation. However, it should be noted that increasing the number of partitions beyond this might result in partitions without objects, causing instability in the formation of ECDF. Therefore, an increase in the number of partitions needs to be carefully managed to avoid instability in ECDF analysis.

The previous research utilizing the MNIST dataset for handwritten digits has implemented various classification techniques, including linear classification with error rates ranging from 7.6% to 12%, K-NN approach with error rates ranging from 1.5% to 5%, non-linear classification with an error rate of approximately 3.5%, and other methods such as SVM with error rates ranging from 1.6% to 4.7%. Additionally, CNN has been used with error rates between 0.7% and 1.7% [49]. The subsequent progress in research indicates a decrease in error rates. The advantage of the approach we applied is its simplicity and ease of interpretation. Therefore, the symbolic data approach using Gaussian copula can be considered a valid alternative for image classification.

## Conclusion

Based on the analysis and experimental findings, it can be concluded that the Gaussian copula model, utilizing a cumulative distribution function space approach, stands as a viable and effective approach for conducting image classification on the MNIST dataset, demonstrating a high level of accuracy. The Gaussian copula proves its capability to model the interdependence of pixel usage across different image partitions. These partitions are strategically designed to encompass diverse object characteristics within the images. With an increase in the number of partitions, the model becomes adept at capturing finer distinctions in object characteristics, facilitating the differentiation between different numbers. The research successfully achieves its main objective of building a classification model rooted in the Gaussian copula, employing

a cumulative distribution function space approach, thereby enhancing the clarity and efficiency of image classification.

## Acknowledgments

We express our gratitude to everyone who contributed to this paper. Special thanks to the reviewers for their valuable input, and to our editorial team and peers for their support.

## Author Contributions

**Conceptualization:** Sapto Wahyu Indratno.

**Data curation:** Kurnia Novita Sari.

**Formal analysis:** Sri Winarni.

**Funding acquisition:** Sapto Wahyu Indratno.

**Investigation:** Kurnia Novita Sari.

**Methodology:** Sri Winarni.

**Project administration:** Sapto Wahyu Indratno.

**Resources:** Kurnia Novita Sari.

**Software:** Sri Winarni.

**Supervision:** Sapto Wahyu Indratno.

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
