## [Decision Letter · Decision Letter 0]

13 Oct 2023

PONE-D-23-26454Classification of images using Gaussian copula model in cumulative distribution function spacePLOS ONE

Dear Dr. Winarni,

Thank you for submitting your manuscript to PLOS ONE. After careful consideration, we feel that it has merit but does not fully meet PLOS ONE’s publication criteria as it currently stands. Therefore, we invite you to submit a revised version of the manuscript that addresses the points raised during the review process.

In addition to the reviewers comments, the following points are also need attention:

1- Comparison with Recent Works: The manuscript would benefit from a more in-depth comparison with recent works in the field. We encourage you to enhance the literature review section by discussing how your research relates to and advances the current state of the art.

2- Checking Referencing Style: There are inconsistencies and inaccuracies in your referencing style. To ensure proper citation and adherence to our journal's guidelines, please thoroughly review and correct the referencing style throughout the manuscript.

3- Proofediting: Your manuscript exhibits various grammatical and language issues that need to be addressed. To ensure clarity and readability, we strongly recommend a comprehensive language check and proofediting of your manuscript. This will help eliminate typographical errors and improve the overall quality of the writing.

We look forward to receiving your revised manuscript.

Kind regards,

Sadiq H. Abdulhussain, Ph.D.

Academic Editor

PLOS ONE

Journal Requirements:

"This research was funded by ITB's Program of Research, Research and Community Service, and Innovation (PPMI) 2023"

"This research was funded by ITB’s Program of Research, Research and Community 302

Service, and Innovation (PPMI) 2023. "

"This research was funded by ITB's Program of Research, Research and Community Service, and Innovation (PPMI) 2023"

"The authors declare no conflict of interest."

Additional Editor Comments:

In addition to the reviewers comments, the following points are also need attention:

1- Comparison with Recent Works: The manuscript would benefit from a more in-depth comparison with recent works in the field. We encourage you to enhance the literature review section by discussing how your research relates to and advances the current state of the art.

2- Checking Referencing Style: There are inconsistencies and inaccuracies in your referencing style. To ensure proper citation and adherence to our journal's guidelines, please thoroughly review and correct the referencing style throughout the manuscript.

3- Proofediting: Your manuscript exhibits various grammatical and language issues that need to be addressed. To ensure clarity and readability, we strongly recommend a comprehensive language check and proofediting of your manuscript. This will help eliminate typographical errors and improve the overall quality of the writing.

Reviewers' comments:

Reviewer's Responses to Questions

**Comments to the Author**

1. Is the manuscript technically sound, and do the data support the conclusions?

Reviewer #1: Yes

Reviewer #2: Partly

2. Has the statistical analysis been performed appropriately and rigorously? 

Reviewer #1: Yes

Reviewer #2: No

3. Have the authors made all data underlying the findings in their manuscript fully available?

Reviewer #1: Yes

Reviewer #2: Yes

4. Is the manuscript presented in an intelligible fashion and written in standard English?

Reviewer #1: Yes

Reviewer #2: No

5. Review Comments to the Author

Reviewer #1: Authors have addressed the queries raised by the reviewer. But, some typo errors are seen through out the paper. The manuscript need to be thoroughly checked. A block diagram is needed for better understanding by the reader. More explanation is required for the results and discussion. Accuracy may not suitable for a unbalanced data. so authors may some other metric. Some of these metric is mentioned in the below recent work.

Thivya Anbalagan, Malaya Kumar Nath, D. Vijayalakshmi, Archana Anbalagan, "Analysis of various techniques for ECG signal in healthcare, past, present, and future", Biomedical Engineering Advances: Elsevier, vol. 6. November 2023, pp. 100089 (1-28). Accepted: May 2023. https://doi.org/10.1016/j.bea.2023.100089

Some of the recent works are not cited and mentioned below:

Maddali Yaswanth, Navin Infant Raj, Malaya Kumar Nath, “Hopfield neural network for classification of digits”, 2022 IEEE 9th Uttar Pradesh Section International Conference on Electrical, Electronics and Computer Engineering (UPCON-2022), 2-4th December, Organized by Indian Institute of Information and Technology Prayagraj, Uttar Pradesh, India. In: Electrical, Electronics and Computer Engineering. https://doi.org/10.1109/UPCON56432.2022.9986401.

Reviewer #2: The manuscript ``Classification of Images Using Gaussian Copula Model in Cumulative Distribution Space" proposes an image classification technique using Gaussian copulas. In particular, the authors seek the joint distribution through copulas of certain features extracted by first partitioning the images and evaluating the empirical cumulative distribution. They subsequently calculate the Beta cumulative distribution of the transformed random variables (pixels). The authors employ this new dataset for the classification by using the Na\\"{\\i}ve Bayes approach where the probability distributions are Gaussian copulas. The novelty of the work lies in the decomposition of the original dataset, considering only specific features extracted. The rest of the paper explore an approach for the classification already used in the literature in particular with the use of Gaussian copula. There are no references in the paper concernig this works, (see Salinas-Gutiérrez 2011, Sen 2015, Tamborrino 2021). There are typos throughout the paper and grammatical mistakes. For instance, on line 63, on line 81, colons are missing. On line 161, it should be "Gaussian copula" and not "copula Gaussian." On line 165, perhaps "distribution functions" was intended? On line 170, there are no references. On line 176, it should be "conditional probability" instead of "condition of the condition probability." In Formula 26 and the subsequent one, there should be a "$Z$" instead of "$z$." There are other punctuation errors to be revised, and I can't list them all. Therefore, I invite the authors to thoroughly revise the text.

Other major observations include: the reader might find it challenging to understand what "N" refers to, as it is used for both the number of pixels and the number of images used. The algorithm is explained very succinctly (lines 150-157). It is not explained that after the partition, the image is vectorized for the calculation of the empirical cumulative distribution, which is an important aspect to emphasize. On lines 181 and 182, there is a sentence completely disconnected from the rest. Then, the metric used for classification results is introduced without justifying why only one metric was chosen, especially since different metrics are commonly used to demonstrate classification effectiveness. Table 3 could be removed as it does not add significant information. The authors claim that this technique is highly advantageous from a computational perspective. However, there are no tables indicating the overall computational cost or a reference to the code used. Furthermore, there are no fair comparisons with other fundamental classification techniques. I strongly recommend thoroughly reviewing the entire work and addressing the issues I have raised. Otherwise, I would be inclined to rejection of the work.

6. PLOS authors have the option to publish the peer review history of their article (what does this mean?). If published, this will include your full peer review and any attached files.

Reviewer #1: **Yes: **Malaya Kumar Nath

Reviewer #2: No

---

## [Author Response · Author response to Decision Letter 0]

1 May 2024

PONE-D-23-26454

Classification of images using Gaussian copula model in empirical cumulative distribution function space

PLOS ONE

Reviewer 1 Comments:

1. Comment: “Authors have addressed the queries raised by the reviewer. But, some typo errors are seen through out the paper. The manuscript need to be thoroughly checked.”

Response:

Thank you for reviewer’s constructive feedback. We acknowledge the presence of typographical errors in the manuscript. We have carefully reviewed and corrected these issues to improve the overall quality of the paper. We appreciate your diligence in pointing out these errors, and we hope the revised manuscript now aligns better with the expected standards. Reviewer’s feedback is invaluable to us.

2. Comment: “A block diagram is needed for better understanding by the reader.”

Response:

Thank you for reviewer’s suggestion. A comprehensive block diagram illustrating the analytical stages has been included on page 7 to represent the research flow visually. The diagram delineates the primary steps, starting from data preparation for training and testing, through forming ECDF and DFD as random variables in the Gaussian copula model, to the Gaussian copula model formulation and parameter estimation using the IFM method. The subsequent stages involve the classification process and evaluation and testing using specific metrics. We hope this block diagram enhances the reader's understanding of the research process.

3. Comment: “More explanation is required for the results and discussion.”

Response:

Thank you for providing valuable feedback on our manuscript. We have included additional explanations in the results and discussion sections in response to reviewer’s suggestions. In particular, we have conducted a detailed analysis of the Cumulative Distribution Function (CDF) and Empirical Cumulative Distribution Function (ECDF). Results from the Gaussian copula model have been presented in various scenarios, including full data, two partitions, three partitions, and four partitions. In addition, we have provided a comprehensive discussion of the classification results and associated evaluation metrics. We hope these additions contribute to a more thorough understanding of our results.

4. Comment: “Accuracy may not suitable for a unbalanced data. so authors may some other metric. Some of these metric is mentioned in the below recent work.

Thivya Anbalagan, Malaya Kumar Nath, D. Vijayalakshmi, Archana Anbalagan, "Analysis of various techniques for ECG signal in healthcare, past, present, and future", Biomedical Engineering Advances: Elsevier, vol. 6. November 2023, pp. 100089 (1-28). Accepted: May 2023. https://doi.org/10.1016/j.bea.2023.100089.”

Response:

Thank you for reviewer’s valuable feedback. We appreciate reviewer’s concern regarding the suitability of accuracy for unbalanced data. In our study, we used the MNIST dataset, where the number of samples for each class is evenly distributed. We have added the suggested paper as reference material on page 6, lines 236-240:

“Classification performance, in this context, is assessed using accuracy metrics. In addition, precision, recall, and F1 scores are used to comprehensively assess the model's performance. Similar studies also use other metrics for classification evaluation [46]. These evaluation metrics measure image classification's accuracy, completeness, and effectiveness.”

46. Anbalagan T, Nath MK, Vijayalakshmi D, Anbalagan A. Analysis of various techniques for ECG signal in healthcare, past, present, and future. Biomed Eng Adv. 2023 May:100089. https://doi.org/10.1016/j.bea.2023.100089

5. Comment: ”Some of the recent works are not cited and mentioned below:

Maddali Yaswanth, Navin Infant Raj, Malaya Kumar Nath, “Hopfield neural network for classification of digits”, 2022 IEEE 9th Uttar Pradesh Section International Conference on Electrical, Electronics and Computer Engineering (UPCON-2022), 2-4th December, Organized by Indian Institute of Information and Technology Prayagraj, Uttar Pradesh, India. In: Electrical, Electronics and Computer Engineering. https://doi.org/10.1109/UPCON56432.2022.9986401”.

Response:

Thank you for bringing this to our attention. We apologize for our oversight in not explicitly mentioning the papers of Maddali Yaswanth, Navin Infant Raj, and Malaya Kumar Nath. We want to clarify that their work has been cited in our paper, and you can find the relevant citations on page 1.:

Image classification constitutes a fundamental aspect of computer vision, encompassing the systematic categorization of images into distinct classes or categories based on their discernible visual features and characteristics. The modeling process involves acquiring knowledge through pattern learning from the training data. The program's efficacy in capturing these data patterns is directly contingent upon the magnitude of the training dataset [1]. The practical applications of image classification are diverse, spanning tasks such as object identification [2], facial detection [3], handwriting recognition [4–8], automated automotive number plate recognition [9], and medical condition diagnosis [10].

9. Yaswanth M, Raj NI, Nath MK. Hopfield Neural Network for Classification of Digits. 9th IEEE Uttar Pradesh Sect Int Conf Electr Electron Comput Eng UPCON 2022. 2022;1-6. https://doi.org/10.1109/UPCON56432.2022.9986401.

Reviewer 2 Comments:

1. Comment: “The manuscript ``Classification of Images Using Gaussian Copula Model in Cumulative Distribution Space" proposes an image classification technique using Gaussian copulas. In particular, the authors seek the joint distribution through copulas of certain features extracted by first partitioning the images and evaluating the empirical cumulative distribution. They subsequently calculate the Beta cumulative distribution of the transformed random variables (pixels). The authors employ this new dataset for the classification by using the Naive Bayes approach where the probability distributions are Gaussian copulas. The novelty of the work lies in the decomposition of the original dataset, considering only specific features extracted. The rest of the paper explore an approach for the classification already used in the literature in particular with the use of Gaussian copula”.

Response:

We appreciate the thorough evaluation of our manuscript "Classification of Images Using Gaussian Copula Model in Cumulative Distribution Space." The recognition of our proposed image classification technique using Gaussian copulas, especially in exploring joint distribution through copulas of specific features extracted via image partitioning, is duly noted. In response, we will emphasize the distinctive aspects of our approach more clearly in the introduction and methodology sections. Reviewer’s valuable comments are highly appreciated, and we will carefully address any suggestions to enhance the clarity and uniqueness of our research.

2. Comment: “There are no references in the paper concernig this works, (see Salinas-Gutiérrez 2011, Sen 2015, Tamborrino 2021)”.

Response:

Thank you for reviewer’s feedback. We apologize for the oversight. The mentioned references, Salinas-Gutiérrez 2011, Sen 2015, and Tamborrino 2021, have been carefully reviewed and added to the list of references in this paper. It is important to note that these papers are relevant to the work presented in this paper on page 2 lines 57-65.

Several studies have examined applying Gaussian copulas as classification models for image analysis. For instance, a study utilized a Gaussian copula model with the Gamma distribution as the marginal distribution to classify image textures in the Vision Texture database [26]. Another investigation introduced a Gaussian copula-based probabilistic classifier to distinguish between object and background images [27]. Additionally, a research endeavor employed the copula distribution function and a diagnostic classifier to categorize brain image data [28]. The application of Gaussian copulas has extended to statistical pattern recognition for acute inflammation data [29] and the classification of hyperspectral images [30].

27. Salinas-Gutiérrez R, Hernández-Aguirre A, Rivera-Meraz MJJ, Villa-Diharce ER. Using Gaussian Copulas in supervised probabilistic classification. Soft computing for intelligent control and mobile robotics. 2011;318:355–72.

29. Sen S, Diawara N, Iftekharuddin KM. Statistical Pattern Recognition Using Gaussian Copula. J Stat Theory Pract. 2015;9(4):768–77. 

30. Tamborrino C, Mazzia F. Classification of hyperspectral images with copulas. J Comput Math Data Sci. 2023;6(2023):100070. https://doi.org/10.1016/j.jcmds.2022.100070

3. Comment: “There are typos throughout the paper and grammatical mistakes. For instance, on line 63, on line 81, colons are missing. On line 161, it should be "Gaussian copula" and not "copula Gaussian." On line 165, perhaps "distribution functions" was intended? On line 170, there are no references. On line 176, it should be "conditional probability" instead of "condition of the condition probability." In Formula 26 and the subsequent one, there should be a "$Z$" instead of "$z$." There are other punctuation errors to be revised, and I can't list them all. Therefore, I invite the authors to thoroughly revise the text. Other major observations include: the reader might find it challenging to understand what "N" refers to, as it is used for both the number of pixels and the number of images used”.

Response:

We express our gratitude for the thorough review and insightful feedback provided by the reviewer. After carefully considering the suggestions, we have made significant revisions to the manuscript, particularly in terms of writing, aiming to clarify concepts and present our research more clearly. We hope that these changes meet the expectations and standards set forth. Review’s insights are invaluable in improving the manuscript.

4. Comment: “The algorithm is explained very succinctly (lines 150-157). It is not explained that after the partition, the image is vectorized for the calculation of the empirical cumulative distribution, which is an important aspect to emphasize”. 

Response:

Thank you for bringing this to our attention. We appreciate reviewer’s feedback on the algorithm explanation. It is important to note that the algorithm is specifically designed to address parameter estimation in the Gaussian Copula. In response to reviewer’s feedback, we have enhanced the explanation by providing details about the vectorization process after image partitioning to calculate the Empirical Cumulative Distribution Function (ECDF). Please refer to Figure 3 on page 8 for a visual representation of this process.

5. Comment: “On lines 181 and 182, there is a sentence completely disconnected from the rest. Then, the metric used for classification results is introduced without justifying why only one metric was chosen, especially since different metrics are commonly used to demonstrate classification effectiveness.”

Response:

We appreciate the thorough review and comments from the reviewer. The sentence mentioned in lines 181 and 182 has been revised to align with the context of the manuscript. Regarding the evaluation metrics, we have incorporated additional metrics to provide a more comprehensive overview of the model's performance. In this study, model evaluation is conducted using the confusion matrix. Classification performance is assessed using accuracy metrics, and to further enhance the evaluation, precision, recall, and F1 scores are included. These metrics offer a more holistic understanding of the model's performance. We hope these additions strengthen the evaluation and clarity in discussing the model's performance. (page 6, lines 235-240).

6. Comment: “Table 3 could be removed as it does not add significant information. The authors claim that this technique is highly advantageous from a computational perspective. However, there are no tables indicating the overall computational cost or a reference to the code used. Furthermore, there are no fair comparisons with other fundamental classification techniques”.

Response:

We appreciate the thorough feedback from the reviewer. Table 3 presents the covariance matrix values of the Gaussian Copula model parameters for the full dataset. The information obtained from Table 3 indicates that high values suggest a strong relationship between random variables. However, these values show minimal variation across classes, suggesting that distinguishing characteristics between classes cannot be well discerned with the full dataset.

Regarding the computational perspective, we acknowledge that this study does not include an overall analysis of computational costs and code references. The research primarily focuses on implementing the Gaussian Copula model using distribution function features. However, we recognize that a direct comparison with other basic classification techniques and a comprehensive analysis of computational costs, including memory usage, are beyond the scope of this research.

The initial statement in the manuscript regarding memory usage explains that the approach used in this research allows for efficient storage of image features, which take the form of distributions. We only need to store the values of these distributions, eliminating the need to store the entire image data. This approach is expected to save computer memory usage. We appreciate the feedback from the reviewer, and these valuable insights will be considered for future research.

7. Comment: “I strongly recommend thoroughly reviewing the entire work and addressing the issues I have raised. Otherwise, I would be inclined to rejection of the work.”

Response:

We greatly appreciate the thorough evaluation and constructive feedback provided by the reviewer. We are committed to carefully reviewing the entire work and addressing all the issues raised during the revision process to enhance the quality of this manuscript. The reviewer's suggestions are invaluable, and we hope the revised manuscript will meet the expected standards. Thank you for the reviewer's time and consideration.

We commit to revising the manuscript in accordance with these valuable suggestions and feedback. Thank you for your time and consideration.

Best regards,

Dr. Sri Winarni 

Department of Statistics, Faculty of Mathematics and Natural Sciences, Padjadjaran University, West Java, Indonesia

---

## [Decision Letter · Decision Letter 1]

1 Jul 2024

PONE-D-23-26454R1Classification of images using Gaussian copula model in empirical cumulative distribution function spacePLOS ONE

Dear Dr. Winarni,

Thank you for submitting your manuscript to PLOS ONE. After careful consideration, we feel that it has merit but does not fully meet PLOS ONE’s publication criteria as it currently stands. Therefore, we invite you to submit a revised version of the manuscript that addresses the points raised during the review process.

We look forward to receiving your revised manuscript.

Kind regards,

Sadiq H. Abdulhussain, Ph.D.

Academic Editor

PLOS ONE

Additional Editor Comments:

Please Check the attached file for reviewers comments.

Reviewers' comments:

Reviewer's Responses to Questions

**Comments to the Author**

1. If the authors have adequately addressed your comments raised in a previous round of review and you feel that this manuscript is now acceptable for publication, you may indicate that here to bypass the “Comments to the Author” section, enter your conflict of interest statement in the “Confidential to Editor” section, and submit your "Accept" recommendation.

Reviewer #1: All comments have been addressed

Reviewer #2: (No Response)

2. Is the manuscript technically sound, and do the data support the conclusions?

Reviewer #1: Yes

Reviewer #2: Yes

3. Has the statistical analysis been performed appropriately and rigorously? 

Reviewer #1: (No Response)

Reviewer #2: No

4. Have the authors made all data underlying the findings in their manuscript fully available?

Reviewer #1: Yes

Reviewer #2: Yes

5. Is the manuscript presented in an intelligible fashion and written in standard English?

Reviewer #1: Yes

Reviewer #2: No

6. Review Comments to the Author

Reviewer #1: Well organized. The queries raised by the reviewers are well addressed. References are to be in the journal format

Reviewer #2: (No Response)

7. PLOS authors have the option to publish the peer review history of their article (what does this mean?). If published, this will include your full peer review and any attached files.

Reviewer #1: **Yes: **Malaya Kumar Nath

Reviewer #2: No

---

## [Author Response · Author response to Decision Letter 1]

20 Jul 2024

PONE-D-23-26454

Classification of images using Gaussian copula model in empirical cumulative distribution function space

PLOS ONE

Reviewer Comments:

 Comment: “In line 99, the index u of X_(j,u) should be i. In line 131, the letter H is missing, and the index of y should be reviewed because y_ps should be y_kps. In line 139, the partial derivative expression in the numerator should not have the superscripts ps. There are other oversights regarding the indexes, which should be reconsidered for mathematical rigor. Please reread and correct all oversights and inaccuracies throughout the manuscript.Authors have addressed the queries raised by the reviewer. But, some typo errors are seen through out the paper. The manuscript need to be thoroughly checked.”

Response:

Thank you for your valuable feedback. We have reviewed and corrected all the writing errors you mentioned.

Line 99 : The index u of X_(j,u) has been changed to X_(j,i) (line 100).

Line 131 : The missing letter H has been added, and the index of for y has been corrected from y_ps to y_(k,n_p)^p (line 130).

Line 139 : The partial derivative expression in the numerator has been corrected by removing the superscript ps (line 140).

Additionally, we have revised the notation for the indices, moving the indices for partitioning from below to above to facilitate better understanding of the model. For example, the notation y_(k,j,t) which is a random variable on class k, partition j and distinguishing point t is changed to y_(k,t)^j. Then for the number of distinguishing points in each partition which was originally denoted by s the writing was changed to a more general np, adjusting to the number of distinguishing points in the partition. 

We have conducted a thorough review to ensure that there are no other notation and indexing errors throughout the manuscript. All inaccuracies and oversights have been corrected to ensure mathematical rigor. Once again, thank you for your helpful suggestions and comments.

 Comment: “An important issue, Figure 3 represents the quantile function and not the empirical cumulative distribution function (ECDF). Additionally, x ~ must be written correctly and described as vec((A)) ~, because otherwise it is not clear for a reader. It is important to note that the graph shows the quantile function, not the ECDF. Consequently, the graphs in Figure 4 also correspond to quantile functions.A block diagram is needed for better understanding by the reader.”

Response:

Thank you for your valuable feedback. We appreciate the clarity you expect regarding Figure 3 and Figure 4 in our manuscript. We would like to clarify that Figure 3 actually represents the Empirical Cumulative Distribution Function (ECDF), not the quantile function as mentioned. ECDF is an appropriate method to describe the cumulative distribution of pixel values that have been normalized in the range [0, 1]. We will update the paragraph on line 351 to make it clearer. 

The elements of the data matrix A have components x, which is a representation of the pixel intensity values. We convert each x element of the data matrix, initially in the range [0,255], to the range [0,1] by dividing it by 255, expressed as x ~=x⁄255. This transformation results in x ~ values that are easy to handle in numerical calculations, especially in machine learning algorithms and statistical analyses, which are more stable and fast when performed in the interval [0,1]. When the vectorization process converts the data matrix A into a vector, the result is the vector Vec(A). Moreover, the transformation of matrix A yields the matrix A ~, which subsequently transforms into the vector Vec(A ~). Next, the ECDF is constructed by following the method specified in Eq (1). Fig 3 visually illustrates the construction of the ECDF. The ECDF serves as an image characteristic, capturing the pixel values that are less than or equal to x. These characteristics show variations in different image classes, underscoring the importance of identifying specific realization points of x that can effectively distinguish them.

I apologize for any inconvenience, but the notation x ~ in the plots cannot be corrected without rerunning the MATLAB code since the images are not saved.

 Comment: “Figure 5 indicates the quantile function on the side, with the probability density at the bottom and the corresponding cumulative distribution. The letter G indicating the density should be lowercase if the uppercase G indicates the cumulative distribution. These oversights must to be corrected.More explanation is required for the results and discussion.”

Response:

Thank you for your comments and observations. We will address the following points:

a. Figure 5 Clarification: We will ensure that the letter indicating density g is lowercase, while G will denote the cumulative distribution function (CDF). This correction will be implemented to accurately represent both the quantile function and the corresponding probability density.

b. Additional Explanation in Results and Discussion: We recognize the need for more detailed explanation and elaboration in the results and discussion sections. We will provide further clarity to ensure thorough understanding of the presented findings and their implications.

We appreciate your feedback, which will help us enhance the clarity and accuracy of our manuscript.

 Comment: “In Table 2, the parameters for the beta distribution should be labeled as ”beta distribution parameters” instead of ”beta parameters.”

 Response:

Thank you for your comment. We have updated Table 2 to label the parameters for the beta distribution as "beta distribution parameters" instead of "beta parameters" to ensure clarity and accuracy.

 Comment: ”Overall, the idea is original and deserves to be published. However, there are serious mathematical inaccuracies that need to be carefully reviewed. The topic of copulas is not trivial, and I understand that adapting the formulas for the specific case is very difficult.

Once all these issues are addressed, I can reconsider the manuscript for publication.”

Response:

Thank you for your thoughtful review and feedback. We appreciate your recognition of the originality of our work and your consideration for publication. We recognize the importance of addressing mathematical inaccuracies and ensuring the correctness of formulas, especially in adapting them for the specific case of copulas. 

We are committed to thoroughly reviewing and correcting all identified issues to meet the standards required for publication. Your comments will guide us in making the necessary improvements, and we look forward to resubmitting the manuscript for your reconsideration once the revisions are completed.

We commit to revising the manuscript in accordance with these valuable suggestions and feedback. Thank you for your time and consideration.

Best regards,

Dr. Sri Winarni 

Department of Statistics, Faculty of Mathematics and Natural Sciences, Padjadjaran University, West Java, Indonesia

---

## [Decision Letter · Decision Letter 2]

21 Aug 2024

Classification of images using Gaussian copula model in empirical cumulative distribution function space

PONE-D-23-26454R2

Dear Dr. Winarni,

We’re pleased to inform you that your manuscript has been judged scientifically suitable for publication and will be formally accepted for publication once it meets all outstanding technical requirements.

Kind regards,

Sadiq H. Abdulhussain, Ph.D.

Academic Editor

PLOS ONE

Additional Editor Comments (optional):

Reviewers' comments:

Reviewer's Responses to Questions

**Comments to the Author**

1. If the authors have adequately addressed your comments raised in a previous round of review and you feel that this manuscript is now acceptable for publication, you may indicate that here to bypass the “Comments to the Author” section, enter your conflict of interest statement in the “Confidential to Editor” section, and submit your "Accept" recommendation.

Reviewer #1: All comments have been addressed

Reviewer #2: All comments have been addressed

2. Is the manuscript technically sound, and do the data support the conclusions?

Reviewer #1: Yes

Reviewer #2: Yes

3. Has the statistical analysis been performed appropriately and rigorously? 

Reviewer #1: Yes

Reviewer #2: Yes

4. Have the authors made all data underlying the findings in their manuscript fully available?

Reviewer #1: Yes

Reviewer #2: Yes

5. Is the manuscript presented in an intelligible fashion and written in standard English?

Reviewer #1: Yes

Reviewer #2: Yes

6. Review Comments to the Author

Reviewer #1: (No Response)

Reviewer #2: I have no further comments to make, I appreciate the authors' efforts in improving the manuscript which I now think is ready for publication.

7. PLOS authors have the option to publish the peer review history of their article (what does this mean?). If published, this will include your full peer review and any attached files.

Reviewer #1: **Yes: **Malaya Kumar Nath

Reviewer #2: No

---

## [Editor Report · Acceptance letter]

27 Sep 2024

PONE-D-23-26454R2 

PLOS ONE

Dear Dr. Winarni, 

I'm pleased to inform you that your manuscript has been deemed suitable for publication in PLOS ONE. Congratulations! Your manuscript is now being handed over to our production team.

Kind regards, 

on behalf of

Dr. Sadiq H. Abdulhussain 

Academic Editor

PLOS ONE